# Long-range correlation in protein dynamics: Confirmation by structural data and normal mode analysis

Qian-Yuan Tang *, Kunihiko Kaneko

Center for Complex Systems Biology, Universal Biology Institute, University of Tokyo, Tokyo, Japan

* tang@complex.c.u-tokyo.ac.jp

## Abstract

Proteins in cellular environments are highly susceptible. Local perturbations to any residue can be sensed by other spatially distal residues in the protein molecule, showing long-range correlations in the native dynamics of proteins. The long-range correlations of proteins contribute to many biological processes such as allostery, catalysis, and transportation. Revealing the structural origin of such long-range correlations is of great significance in understanding the design principle of biologically functional proteins. In this work, based on a large set of globular proteins determined by X-ray crystallography, by conducting normal mode analysis with the elastic network models, we demonstrate that such long-range correlations are encoded in the native topology of the proteins. To understand how native topology defines the structure and the dynamics of the proteins, we conduct scaling analysis on the size dependence of the slowest vibration mode, average path length, and modularity. Our results quantitatively describe how native proteins balance between order and disorder, showing both dense packing and fractal topology. It is suggested that the balance between stability and flexibility acts as an evolutionary constraint for proteins at different sizes. Overall, our result not only gives a new perspective bridging the protein structure and its dynamics but also reveals a universal principle in the evolution of proteins at all different sizes.

**Data Availability Statement:** All the protein structures used in this research are available from the Protein Data Bank (PDB). Related PDB-ID, code, and the data that related to this study are provided as Supporting File.

## Author summary

The long-range correlated fluctuations are closely related to many biological processes of the proteins, such as catalysis, ligand binding, biomolecular recognition, and transportation. In this paper, we elucidate the structural origin of the long-range correlation and describe how native contact topology defines the slow-mode dynamics of the native proteins. Our result suggests an evolutionary constraint for proteins at different sizes, which may shed light on solving many biophysical problems such as structure prediction, multiscale molecular simulations, and the design of molecular machines. Moreover, in statistical physics, as the long-range correlations are notable signs of the critical point, unveiling the origin of such criticality can extend our understanding of the organizing principle of a large variety of complex systems.

**Funding:** This research was partially supported by a Grant-in-Aid for Scientific Research (S) (15H05746) from the Japanese Society for the Promotion of Science (JSPS) and Grant-in-Aid for Scientific Research on Innovative Areas (17H06386) from the Ministry of Education, Culture, Sports, Science and Technology (MEXT) of Japan. The funders had no role in study design, data collection and analysis, decision to publish, or preparation of the manuscript.

**Competing interests:** The authors have declared that no competing interests exist.

## Introduction

Proteins, including the globular, fibrous, membrane and intrinsically disordered proteins, are responsible for diverse functions in almost every process of cellular life. Globular proteins, as the majority type of the proteins in nature, can fold from disordered peptide chains into specific three-dimensional (3D) structures on minimal-frustrated energy landscape [1–4]. Such kind of 3D structures, which are encoded by the amino acid sequences, are known as native states. It is worth noting that the native state of a protein is not static, but exhibits dynamical fluctuations around the energy minimum. Experiments and molecular simulations have shown that thermal fluctuations trigger the motions of proteins such as domain movements and allosteric transitions, which enable the biological functions of proteins such as catalysis [5], ligand binding [6, 7], biomolecular recognition [8], and transportation [9]. Uncovering the relations between the structure and the function of proteins is a fundamental question in molecular biophysics. To answer it, the fluctuations at the native states may provide a key.

One of the most fascinating properties of proteins is the long-range correlated fluctuations around the native states [10–12]. Thanks to the long-range correlations, local perturbations to any residue can be sensed by every other residue of the entire protein, even when the two sites are spatially distant. Such a property plays an important role in the functionality of the proteins. For example, for allosteric proteins, long-range correlations warrant the binding at one site can be transmitted to other functional sites [13, 14], and enable the high susceptibility for proteins in cellular environments. Based on the correlation analysis of structural ensembles determined by solution nuclear magnetic resonance (NMR), it was already demonstrated that the native proteins exhibit long-range correlations and high susceptibility in the native dynamics [15]. Such a phenomenon is also in line with other theoretical and experimental results, for example, the long-range conformational forces related to the hydrophobicity scales of the proteins [16–20], the fractal dimension in the oscillation spectrum [21] and configuration space [22], the slow relaxation of protein molecules in the solution [23, 24], the volume fluctuation of allosteric proteins [25], and the overlap between the low-frequency collective oscillation modes and large-scale conformational changes in allosteric transitions [26–30]. Accumulating evidence indicates that native proteins are not only stable enough to warrant structural robustness, but also susceptible enough to sense the signals in the milieu, and ready to perform large-scale conformational changes. However, the origin of such kind of dynamics is still unclear.

In the present paper, we concentrate on the structure and the equilibrium fluctuation dynamics of a large set of globular proteins determined by X-ray crystallography, ranging from a single hairpin structure to large protein assemblies. Firstly, to elucidate the connection between the long-range correlations and protein structures, we conduct correlation analysis based on the elastic network models (ENMs) [26–30]. We find that the long-range correlations and the scaling laws can be robustly reproduced by the ENMs with different model parameters. Such a result indicates that the long-range correlations are encoded in the native topology of the proteins. Secondly, we conduct normal mode analysis [31–33] for protein molecules, ideal polymer chains, and lattice systems. A similar scaling relation holds for polymers, lattices, and proteins, but the scaling coefficients are different. Such a result shows how native proteins balance between order and disorder, which resemble the physical systems near the critical point of a phase transition. Thirdly, we introduce the average path length and modularity to describe the topological characteristics of the proteins. Scaling relations are also observed between these topological descriptors and the size of the proteins. According to the result of the scaling analysis, we conclude that native proteins show both dense packing and fractal topology. Lastly, we focus on the size dependence of proteins' shape. With a given chain length, the shape of a

protein is not random, but a most-probable shape factor always exists. Such a constraint suggests that native proteins balance between stability and functionality. Overall, our result not only gives a new perspective bridging the protein structure and its dynamics but also reveals a universal principle in the evolution of proteins at all different sizes.

## Results

### The critical dynamics of proteins are robustly encoded in the native structures

In previous studies, based on the structural ensembles determined by solution nuclear magnetic resonance (NMR), it was observed that the native proteins in the solution exhibit long-range correlations and high susceptibility in the dynamics [15]. The native fluctuation of proteins behaves as though they are near the critical point of a phase transition [34–36]. The question arises whether the critical dynamics of native proteins are encoded in the native structure or driven by other factors in the milieu. To answer this question, we employ the minimal model of proteins, the elastic network model (ENM) to conduct our analysis.

In an ENM, a protein molecule is described as a set of nodes (represented by their $C_\alpha$ atoms) connected with edges of elastic springs. As shown in Fig 1A, the 3D structure of a protein can be simplified as a network based on the topology of residue contacts. Note that the elastic networks are constructed only based on the spatial distances between residues. If an ENM can successfully reproduce long-range correlations in the fluctuations of the native proteins, then it can be concluded that the critical dynamics of proteins is encoded by the local contacts in the native structures.

The correlated motions of residues can be represented by a covariance matrix, in which matrix element $C_{ij} = \langle \Delta \vec{r}_i \cdot \Delta \vec{r}_j \rangle$. For simplification, we conduct our analysis based on the Gaussian network model (GNM) [37, 38]. In GNM, the covariance matrix $C$ is proportional to pseudoinverse of the Kirchhoff matrix $\Gamma$, i.e., $C_{ij} = \frac{3k_B T}{\kappa} \cdot [\Gamma^+]_{ij}$ [26, 37]. Normalizing the covariance matrix, a pairwise cross correlation $\phi_{ij} = C_{ij}/\sqrt{C_{ii} C_{jj}}$ an be obtained. Similar to previous works [15, 39, 40], a distance-dependent correlation function $\phi(r)$ can be defined by averaging the correlations for residue pairs at mutual distance $r$, and $\phi(r) = \frac{\sum_{i<j} \phi_{ij} \delta(r - r_{ij})}{\sum_{i<j} \delta(r - r_{ij})}$,

where $r_{ij}$ denote the spatial distance between residue $i$ and $j$, and $\delta(x)$ is the Dirac-delta function selecting residue pairs at mutual distance $r$. Here, the correlation length $\xi$ as the distance where $\phi(r)$ first decays to zero.

To examine whether the correlation scales with the protein size, we sample over the protein data across different sizes. By averaging the distance-dependent correlation function $\phi(r)$ for a subset of proteins, we can define the averaged correlation function $\langle \phi(r) \rangle$ to a group of proteins. Here, we divide the dataset into subsets according to the radius of gyration $R_g$ of the proteins (e.g., subset $\{R_g \sim 12Å\}$ contains proteins at size $11.5Å \leq R_g < 12.5Å$), the distance-dependent correlation functions $\phi(r)$ for proteins at different sizes are calculated. As shown in Fig 1B, the correlation function first decreases from its maximum at short distances, crosses zero at $r = \xi$, continues to decline, reaches a negative minimum. As a notable sign of criticality, for proteins of different sizes, the correlation length $\xi$ is proportional to their radius of gyration $R_g$. Therefore, the correlation functions can be scaled by the size ($R_g$) of the proteins, and all the correlation functions collapse (Fig 1C). This result indicates that correlations in the native fluctuation of proteins are scale-free: No matter how large the protein molecule is, correlation length can extend to the size of the entire system. Such long-range correlation contributes to the functionality of a large variety of proteins, for example, for allosteric proteins, the

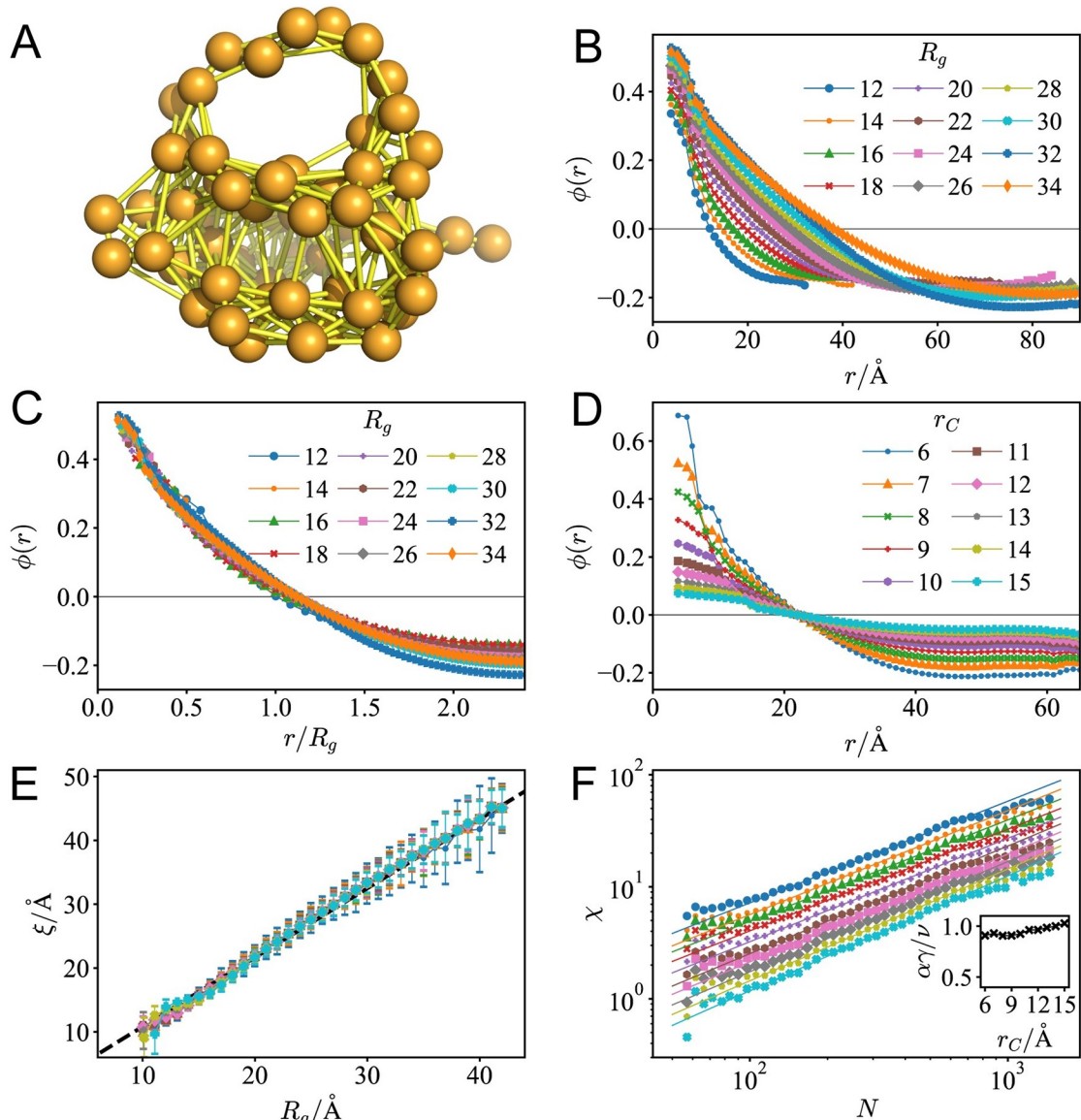

**Fig 1. The critical dynamics of proteins are robustly encoded in the native structure.** (A) An illustration of the elastic network model ($r_C$ = 9Å) of the protein CI2 (PDB code: 2CI2). The beads denote the residues, and the bonds denote the elastic springs in the model. (B) The correlation functions $\phi(r)$ for proteins at different sizes predicted by GNM with cutoff distance $r_C$ = 9Å. (C) Correlation functions scaled by the radius of gyration of the proteins $R_g$. (D) For proteins of similar sizes (19.5Å $\leq R_g <$ 20.5Å), with different cutoff distances $r_C$, the correlation functions $\phi(r)$ predicted by GNM. (E) With different cutoff distances, for proteins of different sizes, the correlation length $\xi$ is always proportional to the size of the protein $R_g$. (F) The susceptibility $\chi$ vs. chain length $N$ shows the power-law relation: $\chi \sim N^{\alpha\gamma/\nu}$, and the scaling coefficient $\alpha\gamma/\nu \approx 1$ can be kept with different $r_C$ (inset).

long-range correlation warrants the binding at one site can be transmitted to other functional sites [13, 14], even when the two sites are spatially distant.

To validate the previous analysis, let us consider the parameter sensitivity in the prediction of the cross correlations in protein dynamics. The only free parameter in GNM is the cutoff distance $r_C$. With different $r_C$, the correlation would have different magnitude at short distances; however, as shown in Fig 1D, the correlation lengths $\xi$ keep as a constant for different cutoff distances $r_C$. As shown in Fig 1E, for cutoff distances ranging from 6 Å to 15 Å, the

correlation length $\xi$ is always proportional to the radius of gyration $R_g$, showing that the critical dynamics of native proteins is generally a stable property and insensitive to the selection of cut-off distances. With only short-range interactions between residues taken into account, GNM can successfully capture the long-range correlations in the native dynamics of the proteins.

To have a further investigation of the criticality, it is necessary to validate the scaling relations in the dynamics of proteins. Here, for illustration, we take the power-law relation between the susceptibility $\chi$ and chain length $N$ as an example. For protein systems, a finite-size version of susceptibility $\chi$ is introduced to quantify the response of systems under perturbation [15]. It is defined as the total correlation in a unit volume within the correlation length: $\chi = \frac{s}{N}\sum_{i<j} \phi_{ij} \cdot \theta(\xi - r_{ij})$, where $s$ denotes the shape factor of protein, and $\theta(x)$ denotes the Heaviside function. Previously, based on NMR-determined protein ensembles [15], it was observed that $\chi \sim N^{\alpha\gamma/\nu}$, with the scaling coefficient $\alpha\gamma/\nu \approx 1$ (Definitions of $\alpha$, $\gamma$ and $\nu$ are listed in S1 Appendix). Here, as shown in Fig 1F, by employing the GNM, similar scaling relations can also be observed. Such a result demonstrates that, no matter how large the molecule is, proteins can always have high sensitivity executing its function because the magnitude of the susceptibility grows with the chain length of the proteins. Besides, the scaling coefficients are insensitive to changes in cutoff distances (inset), demonstrating that the scale-free correlation of native proteins is a robust property.

Our correlation analysis and scaling analysis methods can also be extended to other versions of elastic network models. For example, with harmonic $C_\alpha$ potential model (HCA) [41, 42], similar scaling coefficients can also be observed (see S1 Appendix). However, some models cannot correctly reproduce the scaling relations between $\chi$ and $N$, for instance, the parameter-free GNM (pfGNM) [43]. In fact, pfGNM fails to predict all the scaling relations in the proteins (see S1 Appendix). Previous researches already found that pfGNM can only be applied for proteins in crystalline conditions, and it will have a poor agreement to the collective motions given by molecular dynamics [42]. Such a result indicates that the scaling coefficient may help us to probe whether the protein is solvated or in a crystalline condition.

## The size dependence of slowest modes reveals criticality of native proteins

Normal mode analysis is a practical tool to elucidate the global dynamics [31–33] and the evolutionary constraints [44, 45] of the proteins. Physically, the slow modes, or say, the low-frequency modes of a system are related to the motions with low excitation energy, long wavelengths (long-range correlation), long time scale (at the order from microseconds to seconds) and the large amplitude motions. Usually, the motions that correspond to the slow modes (especially the slowest nonzero mode) can have significant overlap with large displacement during the functional motions [46]. These functional motions usually engage relative movements of large subunits in the proteins or cooperative conformational changes of the whole proteins. Previously, the unique spectral properties of the residue contact networks have been noticed [47, 48], but the detailed differences have never been examined.

To demonstrate the particularity in the spectrum of proteins, we compare the proteins with ideal polymer chains (detailed information listed in S1 Appendix) and lattice systems. Our analysis focuses on the size dependence of the slow modes. As shown in Fig 2A, for all these systems, the slowest few modes versus the system size $N$ follow power-law distributions. Among these slow modes, we specifically focus on the eigenvalue $\lambda_1$ which corresponds to the slowest nonzero mode. A similar power-law $\lambda_1 \sim N^{-\zeta}$ holds for ideal polymers, lattices, and proteins. However, the scaling coefficients $\zeta$ are different in these systems. As shown in Fig 2A, for ideal polymer chains, the scaling coefficient $\zeta \approx 1.674$. For face-centered cubic (fcc) lattice, by conducting normal mode analysis where atoms are connected by springs with their nearest

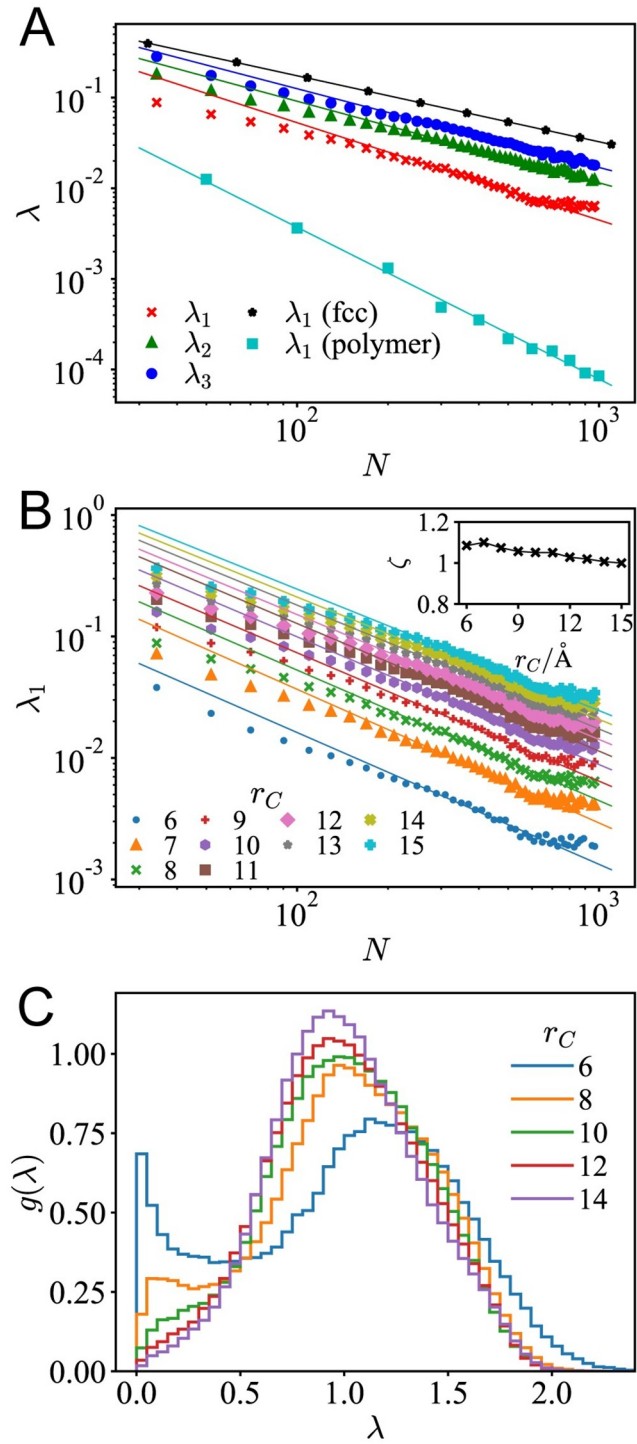

**Fig 2. The slow modes of proteins are robustly defined by native structure.** (A) The 1st, 2nd and the 3rd non-zero eigenvalues $\lambda_1$, $\lambda_2$, and $\lambda_3$ vs. the chain length $N$ of the proteins follows a power-law distribution. (Cutoff distance $r_C$ = 9Å, and the scaling coefficients of $\lambda_1(N)$, $\lambda_2(N)$, and $\lambda_3(N)$ are 1.074, 0.900, and 0.868, respectively). As comparison, similar scaling relations in lattices and ideal polymer chains are also illustrated, and the scaling coefficients are 0.728 (lattices) and 1.674 (polymer). (B) The eigenvalue of the slowest nonzero mode $\lambda_1$ versus chain length $N$ shows the scaling relation: $\lambda_1 \sim N^{-\zeta}$, and the inset shows scaling coefficient $\zeta$ vs. the cutoff distance $r_C$. (C) For proteins at similar sizes (chain length $180 \leq N < 220$), the histogram for the eigenvalue distribution $g(\lambda)$.

neighbors and 2nd nearest neighbors), we have $\zeta \approx 0.727$. Theoretically, for lattice systems, the maximum wavelength $l_w$ corresponds to the slowest elastic mode, and $l_w$ is proportional to the characteristic length of the system. Since the maximum wavelength $l_w \sim N^{1/3}$, one can estimate that $\lambda_1 \sim \omega_1^2 \sim l_w^{-2} \sim N^{-2/3}$, which is close to 0.727. In contrast to ideal polymers and lattices, $\zeta \approx 1$ holds for protein molecules.

The scaling relations in the slowest modes of proteins are robust to the variation in model parameters. As shown in Fig 2B, the selection of cutoff distances $r_C$ would not affect the scaling coefficient $\zeta$. But the robustness of the scaling coefficient cannot be attributed to that of the eigenvalue distribution. As shown in Fig 2C, selecting different $r_C$ would influence the mode distribution $g(\lambda)$ of native proteins. The mode distribution $g(\lambda)$, especially the low-frequency part, can be enhanced by selecting a short cutoff distance $r_C$. Such a result is also consistent with previous theoretical analysis on protein elastic network and ranges of cooperativity [43], which states that with a shorter interaction range, the predicted dynamics would be more cooperative and show better overlap with the displacement in large-scale conformational changes.

It is worth noting that the scaling coefficients in the size dependence of the slowest mode demonstrate that the structure of proteins stands between lattices and ideal polymer chains. For proteins, the exponent $\zeta \approx 1$, above what is obtained from lattices ($\zeta \approx 0.727$), and below what is obtained from polymer chains ($\zeta \approx 1.674$). Thus, compared with ideal polymer chains, the proteins have higher structural stability, whereas compared with lattices, the proteins have higher flexibility and exhibit slower vibrations. Native proteins stand between lattices and polymers, acting as the "critical point" that separates the ordered and disordered phase. Not only are native proteins stable enough to ensure structural robustness and functional specificity, but also susceptible enough to sense the signals in the environment, and ready to perform large-scale conformational changes. Interestingly, staying at the critical point seems to be a common organizing principle of a large variety of biological systems [49–55]: If the system is too disordered, the system cannot stably exist; if it is too ordered, it cannot adapt or respond to perturbations from the environments. Our result of scaling analysis provides additional evidence to support the criticality hypothesis.

## Protein structure: Dense packing with fractal topology

In previous sections, we demonstrated that the critical dynamics of the proteins are encoded in their native structures, and we showed that the equilibrium dynamics of protein molecules if different from lattices and polymers. How does the topology of the residue contact network encode such kind of dynamics? To answer the question, in this subsection, we will try to bridge the vibration spectrum with the architecture of the protein by mainly focusing on the issue of the network topology.

In the network analysis, the average path length $\langle l \rangle$ is one of the most important topological descriptors quantifying the total connectivity among the nodes. Here, we first focus on the scaling relations between average path length $\langle l \rangle$ and the system size $N$. As shown in Fig 3A, for proteins at different sizes, there is a power-law relation between the average path length $\langle l \rangle$ and the chain length $N$: $\langle l \rangle \sim N^{\alpha}$, and $\alpha \approx 0.338$, which is close to 1/3. In the calculation, the cutoff distance $r_C$ is set to be 8Å. Even different cutoff distance $r_C$ will lead to different $\langle l \rangle$, but the scaling exponent is invariant (see S1 Appendix). The scaling relation in proteins is very similar to what in the lattice structures. Theoretically, for 3D lattices, the exponent would be $\alpha = 1/3$. Such a scaling relation is confirmed in Fig 3A. While for ideal polymer chains, with an extended structure, there would be longer average path lengths, and fitting gives $\alpha \approx 0.675$. Such a result demonstrates that the residue contact networks show similar dense packing

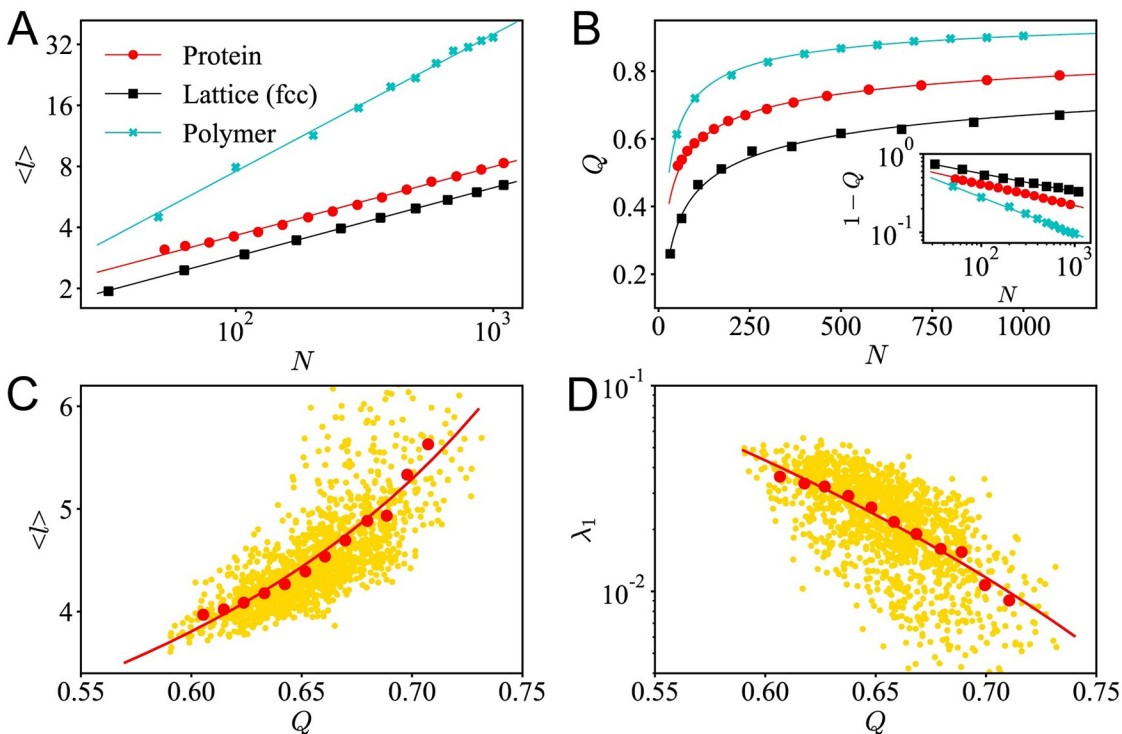

**Fig 3. The protein dynamics can be quantified by topological descriptors of the residue contact network.** (A) For the contact network of proteins ($r_C = 8$Å), fcc lattices and ideal polymers, the average path length $\langle l \rangle$ vs. system size $N$. (B) Similarly for proteins, fcc lattice and ideal polymers, modulaity $Q$ vs. system size $N$. The inset shows the log-log plot of $1 - Q$ vs. $N$. (C) For proteins at similar sizes ($180 \leq N < 220$), the scattering plot (yellow dots, each dot represents a protein molecule), the binned average (red dots) and the basic trend (red curve) of the average path length $\langle l \rangle$ vs. $Q$, and (D) Smallest non-zero eigenvalue $\lambda_1$ vs. $Q$.

property as regular lattices. Both lattice and protein networks have much shorter path length $\langle l \rangle$ than ideal polymers.

Although protein and lattice share similar dense packing properties, the residue contact networks of proteins still exhibit unique properties. To demonstrate the difference between the residue contact network and the lattice networks, another measure—modularity $Q$ is introduced into the study [56, 57]. Intuitively, a network that can be more easily divided into modules would have a higher $Q$ value. Modularity $Q$ also scales as the system size increases. For a $d$–dimensional cubic lattice network with $N$ nodes, theoretically, it was proved that the modularity $Q$ versus $N$ follows the relation: $Q = 1 - K \cdot N^{-\eta}$, where the scaling coefficient $\eta = \frac{1}{d+1}$, and $K$ is a constant that depend on average degree $z$ and dimension $d$ [58]. For ideal polymer chains, the fitting gives $\eta \approx 0.465$, indicating an effective fractal dimension $d_{\text{eff}} \approx 1.15$, which is much lower than 3. For a 3D cubic lattice, theoretically, $\eta = 1/4$. For fcc lattices, as shown in Fig 3B, fitting gives $\eta \approx 0.231 < 1/4$, indicating $d_{\text{eff}} \approx 3.33 > 3$, that is because, in the fcc lattices, every atom has more neighbors than cubic lattice. For proteins our dataset, when taking $r_C = 8$Å, similar power law can also be observed, but the scaling coefficient $\eta = 0.279 > 1/4$. Such an exponent indicate that the proteins has an effective dimension $d_{\text{eff}} = \frac{1}{\eta} - 1 \approx 2.58$, which is lower than 3. Such a scaling coefficient displays that the residue contact networks have a fractal topology, and the fractal dimension is below 3. It is worth noting that, in this work, the fractal dimension of proteins is obtained by the scaling analysis for proteins at different sizes. The effective dimension obtained here is consistent with the fractal dimension ($d \approx 2.7$) of proteins determined by structural analysis methods (see S1 Appendix). The scaling

analysis of average path length reveals that the proteins have similar dense packing properties as ordered lattices, but the scaling analysis of modularity suggests that proteins exhibit fractal structures, which is similar to disordered polymer structures. In short, topological analysis demonstrates again that native of proteins balance between order and disorder.

In the discussions above, by averaging the topological descriptors of proteins at similar sizes, we analyze the size dependence of topological properties. In fact, for proteins at similar sizes, topological descriptors can also play an important role in capturing the main features in the dynamics of the proteins. To illustrate that, here, we select the protein molecules with chain length $180 \leq N < 220$ from our dataset. Although these proteins have similar chain length, the structure may differ a lot. Our discussion centers around modularity $Q$. When the modularity $Q$ of a protein increases, as shown in Fig 3C, the average path length $\langle l \rangle$ also increases. This is because, in a highly modularized network, there will be few connections between different communities, on the average, it will take more steps from one node to another. As shown in Fig 3D, as the modularity $Q$ increases, the smallest non-zero eigenvalue $\lambda_1$ decreases, in line with the common knowledge that that modularized structures in the proteins contribute to slow-mode motions. Such a result is consistent with the theory of spectral graph theory. Indeed, the spectrum of the graph Laplacian is closely related to the community structures of the network [59]. Our analysis quantitatively demonstrates that modularized structures contribute to the large-scale motions and slow relaxations of the proteins.

## Stability-functionality constraint: The size dependence of proteins' shape

The intrinsic dynamics of proteins is encoded in their structures. Since scaling relation between the dynamics and the size of the protein is already discussed in the previous sections. We focus on the relationship between the structure and the size of the protein in this section.

The shape factor $s$ can be introduced to describe the general architecture of a protein molecule [15]. According to the definition, the shape factor can be understood as the residue packing density within the inertia ellipsoid. When residues are tightly packed with a globular shape, the shape factor $s$ would be large. When disordered loops or flexible linkers are connecting multiple domains, the shape of the molecule deviates from an ellipsoid, then $s$ would be small. Here, for illustration, three proteins with a similar chain length $180 \leq N < 220$ but with different shape factor $s$ are shown in Fig 4A. On the left, the receptor-binding domain of the short tail fiber (STF) is illustrated. Such a molecule has hardly any regular secondary structures like $\alpha$−helices or $\beta$-strands [60]. The structure of such a molecule in its monomer state has a small shape factor and high modularity. To perform its functions, a knitted trimeric assembly has to be formed [60]. In the middle, there is the human molecular chaperone heat-shock protein 90 (Hsp90) [61] with medium shape factor and modularity. On the right, a *de novo* designed helical repeat protein DHR10 is illustrated. By repeating a simple helix–loop–helix–loop structural motif, DHR10 protein is highly ordered and becomes very stable, which can stay folded even at 95°C [62]. Generally, the proteins with larger shape factors show higher stability, and the proteins with smaller shape factors show higher flexibility.

Although the definition of shape factor does not introduce any detailed information on secondary structures or residue contacts, the shape factor is closely related to the topological descriptors of the residue contact network. Here, statistics for the proteins with similar chain length ($180 \leq N < 220$) is conducted. The scattering plot of shape factor $s$ versus modularity $Q$ is shown in Fig 4B. A trend line (in red) displays that as modularity $Q$ increases, the shape factor $s$ decreases. The result is easy to understand intuitively, a protein molecule in a shape that deviates from an ellipsoid is likely to have multiple domains or have flexible linkers connecting multiple ordered regions. Interestingly, although the proteins could have very different shapes,

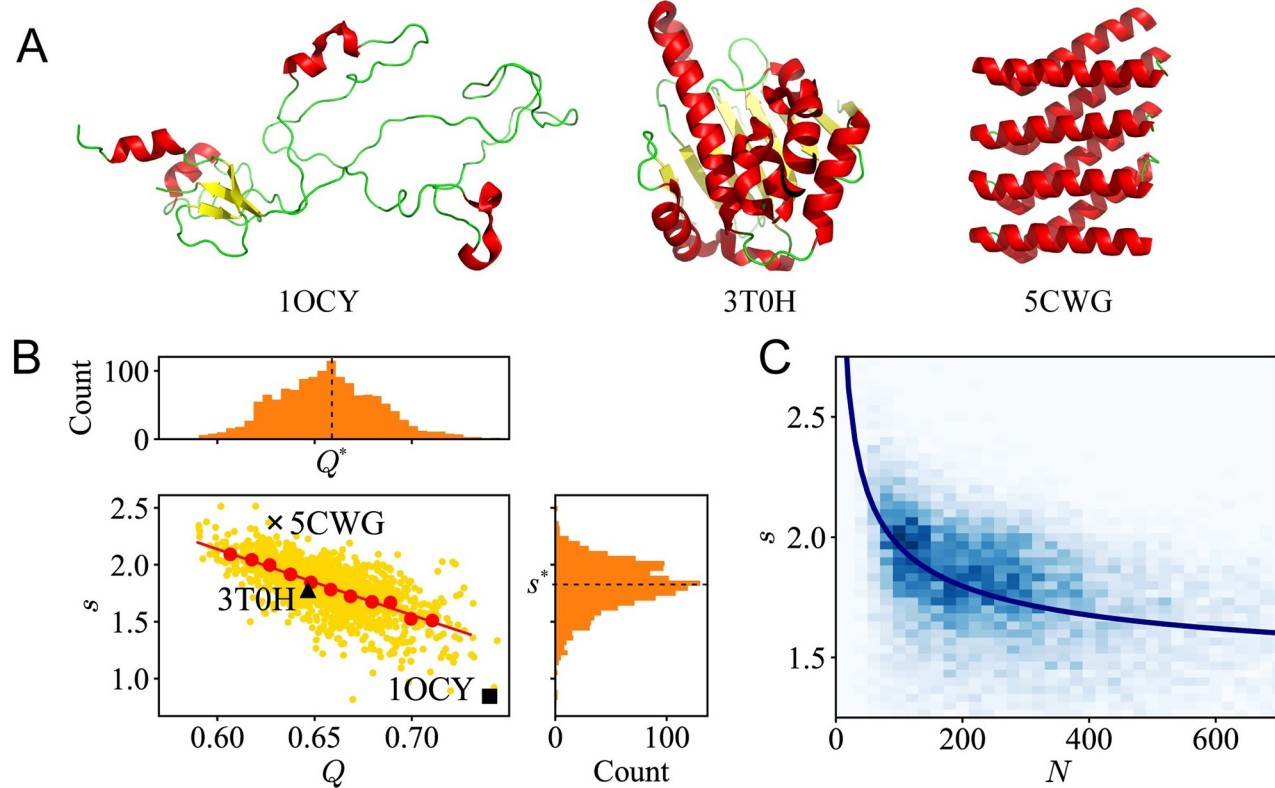

**Fig 4. The shape factor correlates with the chain lengths of the proteins.** (A) Three proteins with similar chain lengths: (Left) The receptor-binding domain of T4 STF (PDB: 1OCY, $s = 0.84$, $Q = 0.74$); (Middle) Human Hsp90 protein (PDB: 3T0H, $s = 1.77$, $Q = 0.65$); and (Right) The DHR10 protein (PDB: 5CWG, $s = 2.37$, $Q = 0.63$). (B) For proteins at similar sizes (chain length $180 \leq N < 220$), the scattering plot (yellow dots), binned average (red dots) and the trend line (red line) of shape factor $s$ vs. modularity $Q$ are plotted. Besides, there are histograms of the shape factor $s$ (right vertical) and modularity $Q$ (top horizontal). (C) For all the proteins in our dataset, the 2D histogram (in the background) of $s$ vs. $N$ and the plot (in navy blue) of the most-probable shape factor $s^*$ vs. chain length $N$.

for protein molecules with a specific chain length, the value of shape factor does not vary a lot. Here, in Fig 4B, histograms of the shape factor $s$ (right vertical) and modularity $Q$ (top horizontal) are plotted. The histograms show that there exists a most-probable shape factor $s^*$ and corresponding modularity $Q^*$. Most natural proteins have shape factors close to $s^*$, exhibit a balancing behavior between stability and flexibility [21].

In fact, for proteins with different chain lengths, the most-probable shape factor $s^*$ always exists, which can be recognized as a constraint in the shape of the protein. As shown in Fig 4C, it was observed that larger proteins prefer smaller shape factors. A similar relation is also observed based on NMR-determined ensembles [15]. These observations provide additional pieces of evidence to support the criticality of native proteins. The native proteins have to balance between stability and flexibility. With short chain lengths, the proteins tend to have a larger shape factor to ensure a stable folded state. Accordingly, small proteins usually have higher residue packing density. However, as the chain length of the proteins increases, to execute functional motions, flexibility becomes the main demand of the proteins. One good example is the designed protein DHR10 as illustrated in Fig 4A. DHR10 has high structural stability, but it is hard for such a protein to execute any biological functions. In such a situation, smaller shape factors, which usually correspond with disordered loops or multi-domain structures, are demanded by the functionality. Our results suggest that the balance between stability and flexibility acts as an evolutionary constraint for proteins at different sizes.

## Discussion

The long-range correlated fluctuations contribute to many biological processes of the proteins, such as allostery, catalysis, and transportation. To understand the origin of such long-range correlations, based on the elastic network model, we conduct normal mode analysis for a large dataset of globular proteins determined by X-ray crystallography.

First, we predict the correlated motions for proteins at different sizes. It is observed that the correlation length of a protein can extend to the size of the whole protein, no matter how large the protein molecule is. Moreover, with different model parameters, the scale-free correlations and the scaling laws can be reproduced by the elastic networks model, which is the minimal structure-based model of native proteins. Such a result indicates that the critical dynamics characterized by the power-law relations are robustly encoded in the native topology of the proteins.

Second, for proteins at different sizes, we conduct normal mode analysis and perform scaling analysis for the slow vibration modes of the proteins. To demonstrate the particularity in the spectrum of proteins, we compare the proteins with ideal polymer chains and lattice systems. Native proteins stand between ordered lattices and disordered polymers, acting as the "critical point" that separates the ordered and disordered phase. Our result of scaling analysis provides additional evidence to support the criticality hypothesis.

Third, to understand how the native topology determines the architecture and the dynamics of the proteins, we conduct scaling analysis for the topological descriptors and the size of the proteins. Our results demonstrate that, although proteins have similar average path length with lattice structures, the residue contact networks are more modularized.

Last, we focus on the size dependence of proteins' shape. For proteins with different chain lengths, the most-probable shape factors always exist. Larger proteins prefer smaller shape factors. Such a constraint results from the balance between stability and functionality of proteins.

In summary, our work quantitatively demonstrates how the native contact topology defines the long-range correlations and the slow dynamics of the native proteins. Our work not only provides quantitative scaling relations supporting the "structure-dynamics-function" paradigm but also reveals evolutionary constraints for proteins at different sizes. These results may shed light on a large variety of biophysical problems such as structure prediction, multi-scale molecular simulations, and the design of molecular machines.

## Materials and methods

### Dataset

Our dataset contains 13081 proteins selected from the Protein Data Bank (PDB) [63]. The structures of these proteins are all determined by X-ray diffraction with high resolution ($\leq 2.0$Å). For every protein structure in the dataset, it contains no DNA, RNA or hybrid structures; and the chain length $30 \leq N \leq 1200$. In our protein dataset, every two proteins share less than 30% sequence similarity. The PDB codes of all the proteins in our dataset are listed in the Supplementary Information (S1 and S2 Files).

### The elastic network models

The elastic network models are widely applied to predict the functional dynamics of a variety of proteins and bio-machineries [26, 27, 29, 30]. With the assumption that all residue fluctuations are Gaussian variables distributed around their equilibrium coordinates, the Gaussian Network Model (GNM) can successfully reproduce the residue fluctuations as determined by experiments [37, 38]. For a protein consisting of of $N$ residues, based on the native structure,

the potential energy of the network is given by:

$$V_{GNM} = \frac{\kappa}{2} \sum_{i,j=1}^{N} \Delta \vec{r}_i \cdot \Gamma_{ij} \cdot \Delta \vec{r}_j, \tag{1}$$

in which $\kappa$ is a uniform force constant; $\Delta \vec{r}_i$ and $\Delta \vec{r}_j$ is the displacement of residue $i$ and $j$, respectively; and $\Gamma_{ij}$ is the element of Kirchhoff matrix, or in a graph theory perspective, it is the graph Laplacian of the residue-residue contact network. The elements of matrix $\Gamma$ is defined according to the contact topology of the native structure: for residue pair $i - j$, if $r_{ij} \leq r_C$, then $\Gamma_{ij} = -1$; if $r_{ij} > r_C$, then $\Gamma_{ij} = 0$; and for the diagonal elements, $\Gamma_{ii} = -\Sigma_{j \neq i} \Gamma_{ij} = -k_i$, where $k_i$ denote the degree of node $i$. In GNM with homogenous contact strength, the only control parameter is the cutoff distance $r_C$. With a large $r_C$, residue pairs at long distances can interact with each other; while for smaller $r_C$, only short-range interactions are contributed to the elastic energy of the system. One may also introduce distance-dependent force constants [41–43] to refine the predictions of elastic network models. In these models, the force constants $\kappa_{ij}$ becomes a function of the mutual distance between residue $i$ and $j$. Further details and other variations of the elastic network models are listed in the S1 Appendix.

## Normal mode analysis and the spectrum of the graph laplacian

Based on GNM, by diagonalizing the Kirchhoff matrix $\Gamma$, we can obtain all the eigenvalues and the corresponding eigenvectors describing the motions of every normal mode [32]. To compare the mode distribution for proteins of different chain lengths, the Kirchhoff (Laplacian) matrices correspond to the topology of native proteins are normalized. By normalizing all the diagonal elements as 1, we can obtain the symmetric normalized graph Laplacian [48]:

$$L = D^{-1/2} \cdot \Gamma \cdot D^{-1/2}, \tag{2}$$

in which $D$ is a matrix of all the diagonal elements of matrix $D = \text{diag}[\Gamma_{1,1}, \Gamma_{2,2}, \cdots \Gamma_{N,N}]$, describing the local packing status of each residue. Diagonalizing matrix $L$, then we have $L = U \Lambda U^T$, in which the eigenvalues $\Lambda = \text{diag}[\lambda_0, \lambda_1, \lambda_2, \cdots \lambda_{N-1}]$ ($\lambda_0 \leq \lambda_1 \leq \lambda_2, \leq \cdots \leq \lambda_{N-1}$) and eigenvectors $U = [u_0, u_1, u_2, \cdots u_{N-1}]^T$. The eigenvalue $\lambda_i$ describes the frequency $\omega_i$ of the $i$-th eigenmode ($\lambda_i \sim \omega_i^2$), and the eigenvector $u_i$ describes the motion profile of the corresponding eigenmode. Note that the zero mode corresponds to the eigenvalue $\lambda_0 = 0$, and eigenvector $u_0$ describes the collective translational or rotational motions of the system. The code of normal mode analysis is listed in the Supplementary Code (S2 Appendix and S3 File).

## Shape factor

To have a general description of the structure of a protein molecule, a dimensionless shape factor $s$ is defined [15]. By calculating the the moments of inertia of a protein molecule, one can estimate the residue packing density within the inertia ellipsoid as $s = \frac{Na^3}{L_1 L_2 L_3}$, in which $a = 3.8$Å is the residue size, and $L_1$, $L_2$ and $L_3$ are lengths of the principal axes of the protein ($L_1 > L_2 > L_3$). The shape factors of the proteins in our dataset are listed in the Supplementary Data (S4 File).

## Average path length

The average (or characteristic) path length $\langle l \rangle$ usually works as a measure of the information transfer efficiency on a network. It is defined as the average number of steps along the shortest paths for all possible pairs of network nodes. When $l_{i,j}$ denotes the shortest distance between

node $i$ and $j$, then, the average path length

$$\langle l \rangle = \frac{1}{N(N-1)}\sum_{i \neq j} l_{i,j}.$$

(3)

## Modularity

Modularity is a topological descriptor which is designed to quantify if a network can be easily divided into modules. For a network with $N$ node and $M$ edges, when the topology is described by the adjacency matrix $A$ where $A_{ij} = 1$ if and only if node $i$ and $j$ are connected. Modularity is defined as the fraction of the edges that fall within the given module minus the expected fraction when edges were distributed at random [56, 57]. According to the definition, one can introduce the modularity matrix $B$ with elements $B_{ij} = A_{ij} - \frac{k_i k_j}{2M}$ to describe the expected number of edges between node pairs, in which $k_i$ and $k_j$ denote the degrees of node $i$ and $j$, respectively. Based on matrix $B$, the modularity can be calculated as:

$$Q = \frac{1}{4M}\mathrm{Tr}(\vec{x}^T \cdot B \cdot \vec{x}),$$

(4)

in which $\vec{x}$ is the column vector describing the partition of a network. Vector $x$ has elements $x_i = \pm 1$ indicating the modules to which the node belongs. The value of the $Q$ lies in the range $-1 \leq Q \leq 1$. For any given partition $s$ of a network, one can calculate the $Q$ corresponding to such a partition. The appropriate partition of a network would maximize the modularity $Q$ [64]. In this work, we introduced the Louvain method [65] to partition the network and maximize the value modularity $Q$. The code of topological analysis is listed in the Supplementary Code (S2 Appendix and S3 File).

## Supporting information

**S1 Appendix. Supplementary information.** Detailed descriptions of the structural datasets involved in this research. Additional information concerning the scaling relations, generation of polymer structures, and other variations of elastic network models are also included in the Supplementary Information.
(PDF)

**S2 Appendix. Supplementary code.** The code (written in Python language) for PDB file processing, correlation analysis, normal mode analysis, and topological analysis are listed in Supplementary Code.
(PDF)

**S1 File. The PDB codes and the chain length of the proteins in Dataset A (13081 proteins determined by X-ray crystallography) are listed in the file.**
(TXT)

**S2 File. The PDB codes and the chain length of the proteins in Dataset B (5078 proteins determined by solution nuclear magnetic resonance) are listed in the file.**
(TXT)

**S3 File. A Jupyter Notebook version of the supplementary code.**
(ZIP)

**S4 File. The data (chain length $N$, radius of gyration $R_g$, average path length $\langle l \rangle$, smallest non-zero eigenvalue $\lambda_1$, shape factor $s$ and susceptibility $\chi$) for all the proteins in our**

**dataset are listed in the file.**
(TXT)

## Author Contributions

**Conceptualization:** Qian-Yuan Tang.

**Data curation:** Qian-Yuan Tang.

**Methodology:** Qian-Yuan Tang.

**Supervision:** Kunihiko Kaneko.

**Validation:** Kunihiko Kaneko.

**Writing – original draft:** Qian-Yuan Tang.

**Writing – review & editing:** Kunihiko Kaneko.

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
