## [Decision Letter · Decision Letter 0]

8 Nov 2019

Dear Dr Tang,

Thank you very much for submitting your manuscript 'Long-range Correlation in Protein Dynamics: Confirmation by Structural Data and Normal Mode Analysis' for review by PLOS Computational Biology. Your manuscript has been fully evaluated by the PLOS Computational Biology editorial team and in this case also by independent peer reviewers. The reviewers appreciated the attention to an important problem, but both raised some substantial concerns about the manuscript as it currently stands. While your manuscript cannot be accepted in its present form, we are willing to consider a revised version in which the issues raised by the reviewers have been adequately addressed. We cannot, of course, promise publication at that time.

In particular, the lack of software and data availability is not acceptable. Please follow the guidelines we have published on how to make data and software reproducable, see:

https://journals.plos.org/ploscompbiol/article?id=10.1371/journal.pcbi.1006649

Sincerely,

Bert L. de Groot

Associate Editor

PLOS Computational Biology

Arne Elofsson

Deputy Editor

PLOS Computational Biology

[LINK]

Reviewer's Responses to Questions

**Comments to the Authors:**

Reviewer #1: The paper presents a large-scale analysis of protein dynamics primarily using GNMs. The work appears to have been carried out carefully and the paper is well-written and understandable.

The results reveal a scaling behaviour that appears to be unique to proteins and these results would be of interest to a certain group of scientists although perhaps not many biologists. I recommend that more is written about how these results can impact on our understanding of protein function and how they might help a biologist investigating a particular system.

I appreciate the GNM part of the paper and have no issues with it. However, I do have issues with the part of the paper that using B-factors and indeed I think it is flawed. The authors suggest that those residues with a positive product of Z values are correlated and those with a negative are anticorrelated. They then look at the distance behaviour of the correlation and see that it scales according to the radius of gyration. This shows a "scale-free" behaviour. My interpretation would be different. The correlation distance is not a correlation distance but a distance that relates to a surface boundary region where there is a transition between residues with low B-factors (internal) and to a region where residues have high B-factors (on or near the surface). It is not surprising therefore that this scales with the size of the protein. Can the authors rule out this explanation? If so they should and if not they might consider removing this section and rely instead on the GNM results alone.

I found the paragraph before the Conclusion section long-winded. The authors should consider shortening it.

Reviewer #2: This article presents a network-centric analysis of collective motions in proteins, based on two approaches: (1) the analysis of crystallographic B-factors and (2) the analysis of Elastic Network Models constructed from crystallographic protein structures. They compare protein-derived networks with random networks and interaction networks of lattice structures. Their findings are interesting, shedding new light on phenomena that have been known empirically for a long time. They also seem novel to me, but I may be unaware of similar prior work because I have not followed network-centric approaches specifically over the last few years. For the same reason, I have not been able to verify many specific assertions on network properties made in the article.

My main criticism of this work is that it is not reproducible. No statement is made about the algorithms applied and software used. No references to published software, no project-specific software available as supplementary material. Unfortunately, the PLOS policy on software sharing is weak and unrealistic (https://journals.plos.org/ploscompbiol/s/materials-and-software-sharing), so the lack of software looks compatible wit this policy, but as a reviewer I have to say that it makes it impossible for me to verify the results of this submission.

General comments:

- The authors refer to critical behavior in many places. This term has different (though related) meanings in different disciplines. The authors' use of the term best fits the concept of self-organized criticality in my opinion (which is also suggested by the titles of references 17 to 20). The authors should then say this in the article (or add some other clarifiation if they don't agree with mine). The authors should also explain in much more detail how their work relates to the earlier studies on self-organized criticality in proteins they refer to.

- The choice of random networks and lattice structures as references for comparison looks a bit arbitrary. Random networks do not correspond to any interaction network in physical systems. Lattices do, but they are very far from proteins in terms of physical properties. The most interesting systems to compare to, in my opinion, are non-biological soft matter systems, such as polymers.

- The authors make specific predictions, e.g. on the scaling of the slowest modes with protein size, that should be amenable to experimental validation. Have they searched for experimental studies in the literature?

Specific comments by page number:

Page 2: "Although there are only short-range physical interactions among the residues,"

Residues being charged, there are long-range interactions as well. It is true however that models that leave out the long-range interactions (e.g. ENMs) also exhibit long-range correlations, suggesting that the long-range interactions are perhaps not essential.

Page 3: "B-factors"

The authors repeat a popular false assumption in the study of protein dynamics: the idea that B-factors measure thermal fluctuations. In crystallographic structure refinement, a distribution of conformations is fitted to the observed Bragg peaks. This distribution is most commonly described by a Gaussian model, consisting of an average structure and a variance. The variance matrix is usually approximated by a diagonal matrix, whose non-zero elements are the B-factors. B-factors thus measure all deviations from an ideal crystal at temperature zero: finit-size effects, crystal disorder, and thermal fluctuations. Crystallography studies of the same protein at different temperatures (e.g. PDB codes 1IEE and 2LYM, both for tetragonal lysozyme) show that the influence of temperature on B-factors is very small, suggesting that the dominant contribution is crystal disorder.

This doesn't invalidate the authors' analysis, as crystal disorder effects also spread through the protein via inter-residue contacts. It is only the presentation in terms of conformational fluctuations that needs to be revised.

Page 4: "X-ray diffraction can only provide one static structure"

See above. An X-ray structure is not static, it is the average structure in a Gaussian ensemble.

Page 5: "From the B-factor profile, one can estimate protein flexibility, "

This looks dubious. As said above, B-factors don't really measure fluctuations. And even if they did, it is not obvious how fluctuations at the atomic level, without inter-atomic conformational correlations, can be translated into some measuer of flexibility.

Page 5: "A positive value of C^{(Z)}_ij" implies that the fluctuations of residues i and j are both above or below the average..."

Why is this relevant? Does the average have any scientific interpretation that makes above/below average meaningful?

Page 7: "it can be concluded that the collective motions of residues and the critical fluctuations of native proteins are encoded in the native structures"

The encoding of collective motions in the structure is the fundamental hypothesis of Elastic Network Models, so it cannot be concluded from an analysis of their results. The added value contributed by the authors is a more specific characterization of these dynamics in comparison to other types of networks.

Page 8: "Slow modes and critical fluctuations of native proteins"

The authors use the simplest form of ENM, in which the force constant for each residue pair can only take the values 0 or 1. It has been shown in the past (see e.g. http://doi.org/10.1021/ct400399x) that these models describe the "real" protein dynamics (obtained from experiment or from more detailed models) rather badly and that ENMs with distance-dependent force constants yield better matches. This raises the question if the mode frequency scaling behavior observed by the authors also holds if more accurate protein models are used.

Page 9: "Noting that the dynamics are encoded in the structures, that is to say, the structures of proteins, which are optimized through the process of molecular evolution, are significantly different from the regular 3D lattice structures."

If the goal is to show the difference between biologically evolved systems and simpler non-adaptive physico-chemical systems, the comparison should not be 3D lattices but non-biological soft matter, e.g. polymers.

**Have all data underlying the figures and results presented in the manuscript been provided?**

Reviewer #1: Yes

Reviewer #2: Yes

PLOS authors have the option to publish the peer review history of their article (what does this mean?). If published, this will include your full peer review and any attached files.

Reviewer #1: No

Reviewer #2: Yes: Konrad Hinsen

---

## [Decision Letter · Decision Letter 1]

21 Jan 2020

Dear Dr. Tang,

We are pleased to inform you that your manuscript 'Long-range Correlation in Protein Dynamics: Confirmation by Structural Data and Normal Mode Analysis' has been provisionally accepted for publication in PLOS Computational Biology.

Before your manuscript can be formally accepted you will need to complete some formatting changes, which you will receive in a follow up email. A member of our team will be in touch within two working days with a set of requests.

Best regards,

Bert L. de Groot

Associate Editor

PLOS Computational Biology

Arne Elofsson

Deputy Editor

PLOS Computational Biology

Reviewer's Responses to Questions

**Comments to the Authors:**

Reviewer #1: The authors have addressed my original concerns.

Reviewer #2: In their revision, the authors have taken all of my criticism and suggestions into account, even beyond my expectations. I find the comparison with polymer chains particularly interesting, and the supplied code, although only partial, to be particularly helpful.

**Have all data underlying the figures and results presented in the manuscript been provided?**

Reviewer #1: Yes

Reviewer #2: No: References to experimental input data (PDB entries) have been provided. Generating the figures from this input requires non-trivial software, which is only partially provided. It is not possible for readers to recompute the figures, but it is possible to understand the ideas and models implemented.

PLOS authors have the option to publish the peer review history of their article (what does this mean?). If published, this will include your full peer review and any attached files.

Reviewer #1: No

Reviewer #2: Yes: Konrad Hinsen

---

## [Editor Report · Acceptance letter]

4 Feb 2020

PCOMPBIOL-D-19-01780R1 

Long-range Correlation in Protein Dynamics: Confirmation by Structural Data and Normal Mode Analysis

Dear Dr Tang,

I am pleased to inform you that your manuscript has been formally accepted for publication in PLOS Computational Biology. Your manuscript is now with our production department and you will be notified of the publication date in due course.

With kind regards,

Laura Mallard
